# Prevalence of and risk factors for depression, anxiety, and stress in non-hospitalized asymptomatic and mild COVID-19 patients in East Java province, Indonesia

Michael Austin Pradipta Lusida[1,2‡], Sovia Salamah[3,4‡]*, Michael Jonatan[1,5], Illona Okvita Wiyogo[1,6], Claudia Herda Asyari[1,7], Nurarifah Destianizar Ali[1], Jose Asmara[1,8], Ria Indah Wahyuningtyas[1,9], Erwin Astha Triyono[1,2], Ni Kadek Ratnadewi[1,10], Abyan Irzaldy[11], Firas Farisi Alkaff[4,12]*

1 Indrapura Emergency Field Hospital, Surabaya, Indonesia, 2 Department of Internal Medicine, Faculty of Medicine Universitas Airlangga–Dr. Soetomo General Hospital, Surabaya, Indonesia, 3 Department of Public Health and Preventive Medicine, Faculty of Medicine Universitas Airlangga, Surabaya, Indonesia, 4 Division of Nephrology, Department of Internal Medicine, University Medical Center Groningen, Groningen, The Netherlands, 5 Department of Cardiology and Vascular Medicine, Faculty of Medicine Universitas Airlangga–Dr. Soetomo General Hospital, Surabaya, Indonesia, 6 Department of Pediatric Surgery, Faculty of Medicine Universitas Airlangga–Dr. Soetomo General Hospital, Surabaya, Indonesia, 7 Department of Pulmonology and Respiratory Medicine, Faculty of Medicine Brawijaya University–Dr. Saiful Anwar General Hospital, Malang, Indonesia, 8 Department of Internal Medicine, Faculty of Medicine, Syiah Kuala University—Dr. Zainoel Abidin General Hospital, Banda Aceh, Indonesia, 9 Department of Ophthalmology, Faculty of Medicine Universitas Airlangga—Dr. Soetomo General Hospital, Surabaya, Indonesia, 10 Health Office of Indonesian Navy Fifth Main Base, Surabaya, Indonesia, 11 Karolinska Institute, Solna, Sweden, 12 Division of Pharmacology and Therapy, Department of Anatomy, Histology, and Pharmacology, Faculty of Medicine Universitas Airlangga, Surabaya, Indonesia

‡ MAPL and SS are shared first authors on this work.
* sovia.salamah@fk.unair.ac.id (SS); firasfarisialkaff@fk.unair.ac.id (FFA)

## Abstract

### Background

Despite abundant data on mental health during the COVID-19 pandemic, 3 important knowledge gaps continue to exist, i.e., 1) studies from low-/middle income countries (LMICs); 2) studies in the later period of the COVID-19 pandemic; and 3) studies on non-hospitalized asymptomatic and mild COVID-19 patients. To address the knowledge gaps, we assessed the prevalence of and the risk factors for mental health symptoms among non-hospitalized asymptomatic and mild COVID-19 patients in one LMIC (Indonesia) during the later period of the pandemic.

### Methods

This cross-sectional study was conducted in September 2020 in East Java province, Indonesia. Study population consisted of non-hospitalized asymptomatic and mild COVID-19 patients who were diagnosed based on reverse transcriptase-polymerase chain reaction results from nasopharyngeal swab. Mental health symptoms were evaluated using the Depression Anxiety Stress Scale-21.

**Data Availability Statement:** All relevant data are within the paper and its Supporting Information files.

**Funding:** The authors received no specific funding for this work.

**Competing interests:** The authors have declared that no competing interests exist.

## Results

From 778 non-hospitalized asymptomatic and mild COVID-19 patients, 608 patients were included in the analysis. Patients' median age was 35 years old and 61.2% were male. Of these, 22 (3.6%) reported symptoms of depression, 87 (14.3%) reported symptoms of anxiety, and 48 (7.9%) reported symptoms of stress. Multivariate logistic regression analysis showed that females were more likely to report symptoms of stress (adjusted odds ratio (aOR) = 1.98, p-value = 0.028); healthcare workers were more likely to report symptoms of depression and anxiety (aOR = 5.57, p-value = 0.002 and aOR = 2.92, p-value = 0.014, respectively); and those with a recent history of self-quarantine were more likely to report symptoms of depression and stress (aOR 5.18, p = 0.004 and aOR = 1.86, p = 0.047, respectively).

## Conclusion

The reported prevalence of mental health symptoms, especially depression, was relatively low among non-hospitalized asymptomatic and mild COVID-19 patients during the later period of the COVID-19 pandemic in East Java province, Indonesia. In addition, several risk factors have been identified.

## Introduction

Since 11 March 2020, severe acute respiratory syndrome coronavirus-2 that causes coronavirus disease 2019 (COVID-19) has been classified as a pandemic by the World Health Organization [1]. Currently, this virus has infected more than 500 million people and cause over 6 million deaths worldwide [2]. As human-to-human transmission occurs upon close contact with an infected person via respiratory droplets or aerosols [3], various preventive public health measures such as quarantine, social distancing, curfews, and lockdowns were implemented to prevent the spread of infection [4].

While these measures were deemed effective in limiting progression of the pandemic, they were not without consequences. People had to abruptly change their daily routines, working models, and social interactions. For example, working parents had to also mind their child (ren) while working from home, all meetings had to be switched from offline to online, business and leisure trips had to be cancelled, and physical contact such as handshakes or hugs were even prohibited. Hence, an increase in the prevalence of individuals with mental health symptoms was to be expected [5–7].

In the beginning of the COVID-19 pandemic, only little attention was paid to the impact of the COVID-19 pandemic on mental health [8]; but now, a great number of studies on this topic have been published. In a recent meta-review of meta-analyses, the prevalence of depression and anxiety during the COVID-19 pandemic was reported to be 26.93% and 27.77%, respectively [9]. These values are strikingly higher compared to pre-pandemic era, where the prevalence was estimated to be 4.4% for depression and 3.6% for anxiety [10]. Additionally, several risk factors that could adversely affect mental health during the COVID-19 pandemic have been identified, such as age, gender, educational background, socioeconomic status, marital status, the presence of children, occupation as healthcare worker (HCW), and a history of self-quarantine [11–13]. Nevertheless, despite the growing body of scientific literature on mental health during the COVID-19 pandemic, 3 important knowledge gaps exist.

First, although the COVID-19 pandemic has affected all countries around the globe, its impact on mental health varies across countries, with mental health symptoms being more prevalent in low-/middle-income countries (LMICs) compared to high-income countries [14]. Even so, most of the studies that have evaluated the impact of mental health during the COVID-19 pandemic originated from China, while studies from LMICs are lacking [15–17]. Second, the impact of the COVID-19 pandemic on mental health differs across time periods, with mental health symptoms being more severe in the beginning of the pandemic and become significantly milder in the following months [18]. Nonetheless, majority of the studies that evaluated mental health were conducted in the beginning of the pandemic [16, 18, 19], which may overestimate the magnitude of mental health problems during the COVID-19 pandemic. Third, previous studies have shown that adverse mental health symptoms are more prevalent among COVID-19 patients compared to the general population or HCWs [12, 14, 16, 20, 21]. Even so, studies assessing mental health during the COVID-19 pandemic rarely focused on the COVID-19 patients [22]. Among COVID-19 patients, those who are hospitalized are at higher risk of having adverse mental health symptoms compared to those who are not hospitalized [23–25]. This is because most of the hospitalized patients are patients with severe COVID-19 symptoms [26], and patients with severe COVID-19 symptoms are more likely to have adverse mental health symptoms than the non-severe one [27, 28]. Nevertheless, majority of COVID-19 patients are asymptomatic or presented with only mild symptoms that do not require hospitalization [24, 25, 29–32]. However, data pertaining to the mental health condition of non-hospitalized asymptomatic and mild COVID-19 patients are lacking [33]. In addition, no study has explored the risk factors of adverse mental health symptoms in this group of patients.

Thus, to address the aforementioned knowledge gaps, we conducted a study in Indonesia, one of the LMICs in Southeast Asia, during the later period of the COVID-19 pandemic, with non-hospitalized asymptomatic and mild COVID-19 patients as the study population.

## Materials and methods

### Study design and study population

This cross-sectional study was conducted between 1 and 30 September 2020 in East Java province, Indonesia. During this period, the number of cases was the highest in the country since the beginning of the pandemic [34]. Among 34 provinces in Indonesia, East Java province was the province with the second highest confirmed cases and the highest mortality rate [35].

The study population consisted of non-hospitalized asymptomatic and mild COVID-19 patients. Respondents were recruited at the Indrapura Emergency Field Hospital, the largest government-owned quarantine facility in East Java province, Indonesia. To be admitted to this quarantine facility, patients had to fulfill the following criteria: 1) Tested positive for COVID-19 on reverse transcriptase-polymerase chain reaction (RT-PCR) test; 2) Asymptomatic or having mild COVID-19 symptoms; and 3) able to take care of themselves.

To avoid the possible effect of facilitated quarantine on mental health, respondents were recruited before they underwent quarantine. When the patients came to the registration desk, those who fulfilled the inclusion criteria were verbally offered to participate in this study by the registrar. If the patients agreed to participate, they were asked to sign the informed consent and hand-filled the questionnaire on the spot.

Inclusion criteria for this study were as follows: age ≥ 18 years old, no history of mental illness, able to read and understand Bahasa Indonesia. History of mental illness among prospective respondents was ascertained by the registrar by verbally asking them if they had ever been diagnosed with mental illness or any mental health problems in the past.

## Ethics approval

This study was conducted in accordance with the guidelines in the Declaration of Helsinki and was approved by the ethical review board of the Faculty of Medicine, Universitas Airlangga prior to study initiation (approval number: 201/EC/KEPK/FKUA/2020; approval date: 19 August 2020). All respondents provided written informed consent prior to their inclusion in the study, and information about the study was given before the consent form was signed. Details that might disclose the identity of the respondents were omitted.

## Research instrument

The research instrument used in this study was a questionnaire requesting data on sociodemographic characteristics and responses to the Indonesian version of the Depression Anxiety Stress Scale-21 (DASS-21). Collected sociodemographic data were age, gender, education background, job status, marital status, number of children, and recent self-quarantine history. As this study also aimed to explore the possible risk factors of adverse mental health symptoms, the gathered sociodemographic data were based on literature concerning possible sociodemographic risk factors associated with mental health symptoms during the COVID-19 pandemic [20, 21]. However, since we want to avoid low response rate, we restricted the collection of sociodemographic data to the one that were easily and commonly gathered. DASS-21 is a self-report instrument for evaluating adverse mental health symptoms, which consists of 21 items that assess 3 components, namely symptoms of depression, anxiety, and stress. There are 7 questions for each component, and each question is scored on a 4-point Likert-scale, ranging from 0 (did not apply to me at all / never) to 3 (applied to me very much / almost always). The final score of each component is calculated by multiplying it by a factor of 2. The minimum final score is 0, and the maximum score is 42 for each component. Based on the total score of each component, the responses are categorized as normal, mild, moderate, severe, and extremely severe [36]. The categorization of each component, including the score range, is presented in Table 1. The Indonesian version of DASS-21 had been validated previously and showed good convergence, discriminant validity, and internal consistency (Cronbach's alpha of 0.895) [37].

## Statistical analysis

Data were analyzed using SPSS Statistic for Windows, version 25.0 (IBM Corp., Armonk, N. Y., USA). Data normality was evaluated using one-sample Kolmogorov-Smirnov test and was presented as mean ± standard deviation (SD) for normally distributed data, median [interquartile range (IQR)] for skewed data, and frequency (percentage) for nominal data. To identify the risk factors for depression, anxiety, and stress, two steps logistic regression analysis was used. First, univariate regression was used to identify each sociodemographic variable that was associated with depression, anxiety, and stress. Variables with p-value < 0.25 [38] were then subjected to multivariate regression using backward selection method. Variables with p-value < 0.05 from the multivariate regression analysis were considered as independent risk factors. During the logistic regression analysis, variables with missing data of more than 20%

**Table 1. Categorization and score range of DASS-21 [36].**

|  | Normal | Mild | Moderate | Severe | Extremely severe |
|---|---|---|---|---|---|
| Depression | 0–9 | 10–12 | 13–20 | 21–27 | 28–42 |
| Anxiety | 0–6 | 7–9 | 10–14 | 15–19 | 20–42 |
| Stress | 0–10 | 11–18 | 19–26 | 27–34 | 35–42 |

were excluded, and depression, anxiety, and stress variables were re-categorized as dichotomous (normal or not) variables with the cut-off scores as follows: 9 for depression, 6 for anxiety, and 10 for stress [36].

## Results

From 778 non-hospitalized asymptomatic and mild COVID-19 patients who came to Indrapura Emergency Field Hospital during the study period, 763 participants fulfilled the inclusion criteria to be enrolled in the study. Of them, 608 patients were included in the analysis (79.7% response rate) (Fig 1). Sociodemographic characteristics of the study participants are presented in Table 2.

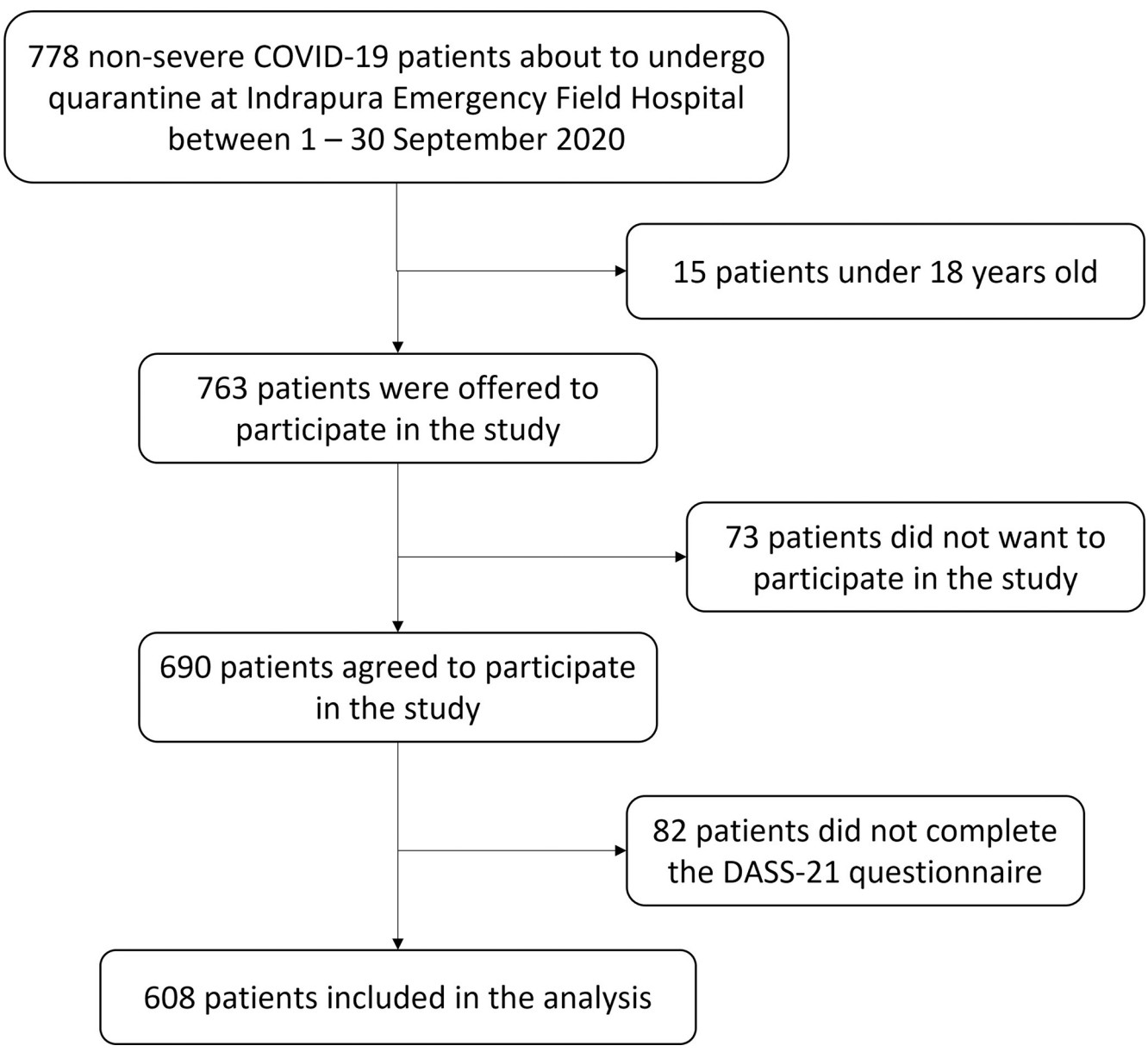

**Fig 1. Flow diagram of study participants.**

**Table 2. Sociodemographic characteristic of the study participants.**

| Variables | N = 608 |
|---|---|
| Age in years, median [IQR] | 35 [27–45] |
| Gender, n (%) | |
| Male | 372 (61.2) |
| Female | 236 (38.8) |
| Marital status, n (%) | |
| Married | 437 (71.9) |
| Single | 162 (26.6) |
| Divorced | 9 (1.5) |
| Have a child, n (%) | |
| Yes | 335 (55.1) |
| No | 273 (44.9) |
| Educational background, n (%)* | |
| Elementary graduate | 16 (3.8) |
| Junior high graduate | 23 (5.4) |
| Senior high graduate | 207 (48.7) |
| University graduate | 179 (42.1) |
| Job status as a healthcare worker, n (%) | 27 (4.4) |
| Family income per month, n (%)# | |
| < IDR 4.000.000 | 168 (41.2) |
| ≥ IDR 4.000.000 | 240 (58.8) |
| Had undergone self-quarantine recently, n (%) | |
| Yes | 271 (44.6) |
| No | 337 (55.4) |

*Missing data on 183 (30.1%) respondents.

#Missing data on 200 (32.9%) respondents.

Median [IQR] score for depression was 0 [0–2], 2 [0–4] for anxiety, and 2 [0–6] for stress. Total score was 4 [0–12]. Of all respondents, 22 (3.6%) reported symptoms of depression, 87 (14.3%) reported symptoms of anxiety, and 48 (7.9%) reported symptoms of stress. Data on the severity of each component is presented in Table 3.

For regression analysis, all sociodemographic variables from Table 1 were included, except educational background and family income, as these variables had a high percentage of missing data. Results of univariate and multivariate regression analysis for depression are presented in Table 4. Respondents who worked as HCWs and those who had undergone self-quarantine recently were more likely to report symptoms of depression (Table 4). Table 5 lists the results of regression analysis for anxiety. Respondents who worked as HCWs were more likely to report symptoms of anxiety (Table 5). Univariate and multivariate regression for stress are

**Table 3. Mental health severity symptoms distribution of the study participants.**

| | Depression | Anxiety | Stress |
|---|---|---|---|
| Normal, n (%) | 586 (96.4) | 521 (85.7) | 560 (92.1) |
| Mild, n (%) | 13 (2.1) | 25 (4.1) | 46 (7.6) |
| Moderate, n (%) | 8 (1.3) | 54 (8.9) | 2 (0.3) |
| Severe, n (%) | 1 (0.2) | 7 (1.2) | 0 (0) |
| Extremely severe, n (%) | 0 (0) | 1 (0.2) | 0 (0) |

**Table 4. Univariate and multivariate logistic regression analysis for depression.**

| Variables | Univariate | | | Multivariate | | |
|---|---|---|---|---|---|---|
| | COR | 95%CI | p-value | AOR | 95%CI | p-value |
| Gender | | | | | | |
| Male (ref) | - | - | - | | | |
| Female | 1.60 | 0.68–3.76 | 0.277 | | | |
| Age in years | 1.01 | 0.97–1.04 | 0.720 | | | |
| Marital status | | | | | | |
| Married (ref) | - | - | - | | | |
| Single | 1.57 | 0.65–3.82 | 0.320 | | | |
| Divorced | 0.0 | 0 | 0.999 | | | |
| Have a child | | | | | | |
| No (ref) | - | - | - | | | |
| Yes | 0.98 | 0.42–2.30 | 0.977 | | | |
| Job status | | | | | | |
| Healthcare workers | 7.54 | 2.55–22.30 | < 0.001 | **5.57** | **1.83–16.95** | **0.002** |
| Other than healthcare workers (ref) | - | - | - | - | - | - |
| Had undergone self-quarantine recently | | | | | | |
| No (ref) | - | - | - | - | - | - |
| Yes | 5.92 | 1.98–17.72 | 0.001 | **5.18** | **1.71–15.69** | **0.004** |

Variables with p-value < 0.25 in univariate analysis were subjected to multivariate analysis. Variables with p-value < 0.05 in multivariate analysis were defined as independent risk factors. AOR, adjusted odds ratio; COR, crude odds ratio; 95%CI, 95% confidence interval.

**Table 5. Univariate and multivariate logistic regression analysis for anxiety.**

| Variables | Univariate | | | Multivariate | | |
|---|---|---|---|---|---|---|
| | COR | 95%CI | p-value | AOR | 95%CI | p-value |
| Gender | | | | | | |
| Male (ref) | - | - | - | | | |
| Female | 1.57 | 1.00–2.48 | 0.052 | | | |
| Age in years | 0.99 | 0.97–1.01 | 0.550 | | | |
| Marital status | | | | | | |
| Married (ref) | - | - | - | | | |
| Single | 0.85 | 0.50–1.45 | 0.554 | | | |
| Divorced | 0.72 | 0.09–5.82 | 0.754 | | | |
| Have a child | | | | | | |
| No (ref) | - | - | - | | | |
| Yes | 1.40 | 0.88–2.23 | 0.159 | | | |
| Job status | | | | | | |
| Healthcare workers | 3.22 | 1.40–7.43 | 0.006 | **2.92** | **1.24–6.88** | **0.014** |
| Other than healthcare workers (ref) | - | - | - | - | - | - |
| Had undergone self-quarantine recently | | | | | | |
| No (ref) | - | - | - | | | |
| Yes | 1.56 | 0.99–2.46 | 0.057 | | | |

Variables with p-value < 0.25 in univariate analysis were subjected to multivariate analysis. Variables with p-value < 0.05 in multivariate analysis were defined as independent risk factors. AOR, adjusted odds ratio; COR, crude odds ratio; 95%CI, 95% confidence interval.

**Table 6. Univariate and multivariate logistic regression analysis for stress.**

| Variables | Univariate | | | Multivariate | | |
|---|---|---|---|---|---|---|
| | COR | 95%CI | p-value | AOR | 95%CI | P-value |
| Gender | | | | | | |
| Male (ref) | - | - | - | - | - | - |
| Female | 2.16 | 1.19–3.92 | 0.011 | **1.98** | **1.08–3.64** | **0.028** |
| Age in years | 1.00 | 0.97–1.02 | 0.882 | | | |
| Marital status | | | | | | |
| Married (ref) | - | - | - | | | |
| Single | 0.92 | 0.47–1.82 | 0.808 | | | |
| Divorced | 1.44 | 0.18–11.81 | 0.737 | | | |
| Have a child | | | | | | |
| No (ref) | - | - | - | | | |
| Yes | 1.39 | 0.76–2.56 | 0.284 | | | |
| Job status | | | | | | |
| Healthcare workers | 3.67 | 1.40–9.58 | 0.008 | | | |
| Other than healthcare workers (ref) | - | - | - | | | |
| Had undergone self-quarantine recently | | | | | | |
| No (ref) | - | - | - | - | - | - |
| Yes | 2.01 | 1.10–3.66 | 0.023 | **1.86** | **1.01–3.44** | **0.047** |

Variables with p-value < 0.25 in univariate analysis were subjected to multivariate analysis. Variables with p-value < 0.05 in multivariate analysis were defined as independent risk factors. AOR, adjusted odds ratio; COR, crude odds ratio; 95%CI, 95% confidence interval.

presented in Table 6. Female respondents and those who had undergone self-quarantine were more likely to report symptoms of stress (Table 6).

## Discussion

Our analysis indicates that, in the later period of the COVID-19 pandemic, the prevalence of depression, anxiety, and stress was 3.6%, 14.3%, and 7.9%, respectively, among non-hospitalized asymptomatic and mild COVID-19 patients in the East Java province, Indonesia. Further, we were able to identify job status as HCWs and recent self-quarantine history to be the risk factors for depression, while that for anxiety was job status as HCWs, and those for stress were female gender and recent self-quarantine history.

To the best of our knowledge, there are only 4 studies that evaluate the mental health of non-hospitalized asymptomatic and/or mild COVID-19 patients to this date [33, 39–41]. Guo et al (2020) evaluated the mental health symptoms of mild COVID-19 patients in China and revealed that the prevalence of depression and anxiety were 17.5% and 6.8%, respectively. Additionally, compared to matched normal individuals, total score for depression and anxiety were significantly higher in mild COVID-19 patients [39]. In Korea, the prevalence of depression and anxiety among asymptomatic and mild COVID-19 patients were 10.3–24.3% and 14.9–15.9%, respectively [33, 40]. A study among asymptomatic COVID-19 patients from India showed that the prevalence of depression, anxiety, and stress were 49.4%, 40.9%, and 75.8%, respectively [41].

There are several possible explanations for such differences in terms of prevalence rates between this current study and previous studies. First, while our study was done in September 2020, previous studies were done during the initial stage of the pandemic. A recent meta-analysis of a longitudinal cohort studies showed that the prevalence of adverse mental health

symptoms was the highest during March–April 2020 and decreased significantly afterward [18]. It is because perceived risks on COVID-19 infection and mortality, financial stability, and lifestyle changes rose sharply in the initial stages of the pandemic and declined in the later stages, and these factors were positively associated with changes in mental health symptoms [42]. Second, as the condition of healthcare systems and the government's response to the pandemic differ across countries, the prevalence of depression, anxiety, and stress are likely to be lower in countries where both are adequate [43, 44]. Third, the instrument used to measure mental health status in this study is different from those used previously [33, 39, 40], which would have also contributed to the observed variation. For example, compared to DASS-21, 8-item Patient Health Questionnaire instrument is more likely to classify individuals as having depression, while 7-item General Anxiety Disorder instrument is more likely to classify individuals as having anxiety [45]. However, the best instrument to measure depression, anxiety, and stress remains contentious, and we used the DASS-21 because it can measure depression, anxiety, and stress with the least number of questions and has already been adapted to Bahasa Indonesia.

Several studies have evaluated the prevalence of mental health symptoms during the COVID-19 pandemic in Indonesia using DASS-21. However, all of the studies focused on either general population [46–48] or healthcare workers [49–53], with none on the asymptomatic and/or mild COVID-19 patients. Depending on the study population and data collection period, the prevalence of depression, anxiety, and stress was 8.5–32.6%, 9.3–44.9%, and 2.4–31.8%, respectively [46–53]. Other than study period differences that has been discussed in the paragraph above and the difference in study population, varying prevalence rates may also be explained by differences in data collection methods used. While previous studies collected the data using online survey [46–52], we directly approached potential respondents and asked them to fill in the questionnaire. When collecting data for mental health studies, it has been reported that respondents provide a more negative response in online surveys than in offline surveys [54]. We hypothesized that this may be due to the anonymity associated with online questionnaires, because the respondents believe that their true identity can be fully protected in online but not in offline surveys. The identity issue might also be associated with societal stigma and discrimination toward people with mental health problems, especially in Asian countries, including Indonesia [55–57].

We found that people who worked as HCWs were more likely to report symptoms of depression and anxiety compared to non-HCWs, a finding not consistent with previous reports. For example, a study from China showed that the general population was at greater risk of developing depression and anxiety compared to HCWs [58], and another study from Italy reported that the general population and frontline HCWs were at similar risk of developing depression and anxiety [59]. We posit that disparities in healthcare systems across countries during a pandemic lead to differential impact on the mental health among HCWs [60], and that they are responsible for the observed differences in results.

The capacity of Indonesia's healthcare system and infrastructure is far from adequate to battle the COVID-19 pandemic. Since before the COVID-19 pandemic, there has been a significant shortage of HCWs and their distribution is uneven throughout the country [61]. However, during the COVID-19 pandemic, the shortage of HCWs is aggravated by high mortality among those treating COVID-19 patients [62, 63], resulting in higher workload and longer working hours for the remaining personnel, especially when the number of COVID-19 patients continue to increase. Moreover, similar to other countries, there is a lack of personal protective equipment (PPE) for HCWs on duty in Indonesia, with this shortage being worsened by the panic buying and stockpiling of medical-grade PPE by the public [63, 64]. Other than that, the number of hospitals, bed capacities, and supporting facilities to treat COVID-19

patients such as negative pressure wards and ICU rooms are lacking and also not evenly distributed in Indonesia [61–63]. The lack of facilities putting HCWs in difficult position, where they have to decide to whom the treatments should be given [65]. The above-mentioned issues might explain why HCWs in Indonesia are more prone to adverse mental health symptoms compared to the general population.

In our study, we discovered that women were more likely to report symptoms of stress, and previous studies, either before [66–69] or during the COVID-19 pandemic [48, 70–72], also showed that women register higher stress scores and are at greater risk of developing stress. Gender differences in mental health have been discussed since the 1970s, and women have been reported to experience distress more frequently and develop more symptoms than men under identical levels of stress [73]. Several explanations have been proposed for greater stress susceptibility in women. Biologically, women express higher levels of corticotropin-releasing factor (CRF) and had more CRF receptors compared to men, and upon its release from the hypothalamus during a stressful event, CRF activates the hypothalamus-pituitary-adrenal axis by stimulating adrenocorticotropic hormone (ACTH) to produce cortisol, which is a primary stress hormone in the body, from the adrenal cortex [74]. Additionally, at identical levels of ACTH, the female adrenal cortex is more responsive to cortisol production than the male adrenal cortex [75]. Furthermore, fluctuations in sex hormones, either due to menstrual cycle or reproductive status, also contribute to stress vulnerability [76]. Psychologically, women tend to express distress by internalizing problems rather than externalizing them [77]. Apart from that, as women are the primary caregivers within the household, and they often prioritize the condition of family members over their own [78, 79]. During the COVID-19 pandemic, it can be assumed that they may be apprehensive of no one being able to take care of the family if they are diagnosed with COVID-19 and had to be quarantined or hospitalized.

Interestingly, although mental health problems appear to be more common in women, suicide rates have been noted to be higher in men [80–82]. It can then be argued that mental health problems are underdiagnosed in men. Several possible explanations have been proposed, and they include: 1) men are less likely to express troubles, discuss sensitive issues, or solve emotional problems [83]; 2) men are more likely to express the distress by externalizing problems rather than internalizing them because of the fear of stigma [77]; and 3) men seek help for mental health care far less often than women as help-seeking behavior is viewed as a weakness and is contrary to masculine traits [84, 85]. In addition to that, it has also been suggested that there is a measurement bias in the currently available self-report instrument for mental health [83]. For instance, the experience of stress is different between sexes, where men feel more depersonalized, while women tend to feel emotionally exhausted [86]. Nevertheless, available instruments to measure stress do not assess psychological stress as depersonalization [68]. Thus, although the current study and a great body of evidence support the notion that women are at higher risk of developing mental health symptoms, we believe that investigations using structured diagnostic interviews should be done in the future to clarify whether one gender is at higher risk of developing mental health symptoms than the other.

In this study, we also found that people who had undergone self-quarantine recently were more likely to report symptoms of depression and stress. Since the 14th century, quarantine has been an important public health measure to reduce incidence and mortality during any outbreak [87, 88]. However, quarantine negatively affects mental health, and data from previous outbreaks have described several adverse psychological effects such as depression, anxiety, stress, low mood, and anger [89]. A recently published meta-analysis revealed a significant relationship between mass quarantine and mental health during the COVID-19 pandemic [90], and a multi-center study from 7 middle-income countries in Asia showed that those who have ever been quarantined during the COVID-19 pandemic were at higher risk of depression,

anxiety, and stress [11]. Studies from China also demonstrate that people who were quarantined had higher risk of depression, anxiety, and stress. Furthermore, those who were diagnosed or suspected of having COVID-19 infection were at even greater risk of depression, anxiety, and stress compared to uninfected individuals [21, 91].

Nevertheless, the negative effects of quarantine on mental health appears to occur only during self-quarantine and not in facilitated quarantine. In the initial stage of the pandemic, Jeong et al (2020) evaluated the mental health of asymptomatic and mildly symptomatic COVID-19 patients who were admitted to the non-hospital facilities for isolation and monitoring in South Korea. Mental health status was evaluated twice in that study, i.e., after the 2nd week of quarantine and 1 week after the first survey, and they found no significant differences in anxiety or depression scores [33]. A similar study from South Korea also found that the prevalence of depression, anxiety, suicidal risk, and stress was constant until the 4th week of quarantine [40]. We have also previously reported that being quarantined in a quarantine facility did not worsen the mental health status of asymptomatic and mild COVID-19 patients [92]. Some of the known psychological stressors during quarantine are frustration, boredom, and inadequate supplies [89], but in a quarantine facility, patients are provided free meals thrice daily, including snacks and entertainment facilities. In contrast, people who undergo self-quarantine are not provided such things by the government, and this might explain these differences in terms of mental health status.

This study has several important limitations. First, the symptoms of depression, anxiety, and stress were based on self-reported questionnaire; hence, they may not always concur with objective assessment by health professionals. Second, the cross-sectional nature of this study precludes any inference of causality or evaluation of longitudinal changes in mental health symptoms during the COVID-19 pandemic. Third, socioeconomic status and educational background were not included in the regression model due to the missing data in more than 30% of the respondents. Fourth, it has been shown that longer quarantine time was associated with worsen mental health status [90, 93]. However, data regarding the number of days in self-quarantine were not available for a majority of the respondents because they could not adequately recall relevant details. Fifth, we did not evaluate the mental health prevalence from other groups, e.g., general population, hospitalized COVID-19 patients, and long COVID-19 patients. Thus, difference in mental health symptoms prevalence between non-hospitalized asymptomatic and mild COVID-19 patients and other groups could not be seen. Last, data collection for this study was done in September 2020. The situation in that period was different than when Indonesia became the epicentrum of the COVID-19 pandemic in Asia [94], or in the recent outbreak of the Omicron variant [95].

## Conclusion

To the best of our knowledge, this is the first study to investigate the mental health symptoms among non-hospitalized asymptomatic and mild COVID-19 patients during the later period of the COVID-19 pandemic. We report that the prevalence of mental health symptoms, especially depression, is relatively low among these patients in East Java province, Indonesia. Despite the low prevalence, our finding showed that HCWs are more vulnerable to depression and anxiety; females are more vulnerable to stress; and those who had undergone self-quarantine recently are more vulnerable to depression and stress.

## Supporting information

**S1 Raw data.**
(SAV)

## Author Contributions

**Conceptualization:** Michael Austin Pradipta Lusida, Sovia Salamah, Michael Jonatan, Firas Farisi Alkaff.

**Data curation:** Sovia Salamah.

**Formal analysis:** Sovia Salamah, Abyan Irzaldy, Firas Farisi Alkaff.

**Investigation:** Michael Austin Pradipta Lusida, Michael Jonatan, Illona Okvita Wiyogo, Claudia Herda Asyari, Nurarifah Destianizar Ali, Jose Asmara, Ria Indah Wahyuningtyas, Erwin Astha Triyono, Ni Kadek Ratnadewi.

**Methodology:** Michael Austin Pradipta Lusida, Sovia Salamah, Michael Jonatan.

**Project administration:** Michael Austin Pradipta Lusida, Michael Jonatan.

**Supervision:** Erwin Astha Triyono, Ni Kadek Ratnadewi.

**Writing – original draft:** Michael Austin Pradipta Lusida, Sovia Salamah, Michael Jonatan, Illona Okvita Wiyogo, Firas Farisi Alkaff.

**Writing – review & editing:** Claudia Herda Asyari, Nurarifah Destianizar Ali, Jose Asmara, Ria Indah Wahyuningtyas, Erwin Astha Triyono, Ni Kadek Ratnadewi, Abyan Irzaldy.

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
