## [Decision Letter · Decision Letter 0]

21 Oct 2021

PONE-D-21-21429Prevalence and risk factor for depression, anxiety, and stress in confirmed non-severe COVID-19 patients in East Java province Indonesia

PLOS ONE

Dear Dr. Salamah,

Thank you for submitting your manuscript to PLOS ONE. After careful consideration, we feel that it has merit but does not fully meet PLOS ONE’s publication criteria as it currently stands. Therefore, we invite you to submit a revised version of the manuscript that addresses the points raised during the review process.

I have read your manuscript with interest and received the comments of two independent reviewers. The comments of the reviewers are attached in this letter. I agree with their very critical comments and will try to prevent repeating them as much as possible. 

My general impression is that a lot of work needs to be done to address the comments of the reviewers and to address my comments. I would like to advise you strongly to ask a native speaker to check/correct the revised manuscript (the language errors distract too much).Your introduction lacks, given your research question, a clear overview of what is (un)known about the mental health of the infected and/or people who are forced to stay in quarantine, as well as groups at risk for mental health problems. It is simply incomplete and too short. Please pay more attention towards the different periods during this pandemic.Please be much more critical on studies examining effects of the pandemic (very often using convenience samples and cross sectional studies without reference data prohibiting any valid conclusion about the effects of this pandemic on mental health), and include the recent meta-analysis of prospective studies of Robinson et al. 2021 (doi: 10.1016/j.jad.2021.09.098).Clarify which IRB was involved (not only the code of 201…..).Please explain how potentially respondents were contacted to ask to participate in your study. How were the questions administered (written, online, verbal?).  When were the questions administered (during quarantine, afterwards and how many weeks after the quarantine). Were other family members infected, in hospital because of infection, deceased following infection, etc.No information is provided about the non-response and results of the non-response analyses.Cronbachs alpha’s of the DASS-21 scales of the study sample are missing.Explain how a history of mental illness was assessed.Explain and clarify which “independent variables” were examined (and introduce/explain alll in the introduction).You entered the variables in the multivariate analyses that were significant in the univariate analyses on p <.025 level. I don’t understand this strategy. It is more informative to include all predictors in the multivariate analyses (and thus you can skip the univariate analyses).Show the prevalence of anxiety, depression and stress of all predictors in the table, and clarify which cut-off’s were used in the logistic regression analyses.The text suggest that you only examined age, gender, marital status, having a child, and income, besides administering the DASS-21. Is this correct? If not, why did you not include these other variables?Please improve your tables and present all relevant info, including explained variance, statistics of all include variables.As the reviewers noted, the discussion is not a discussion (for example a limitation section and comparisons with the results of other studies on infected/quarantined are completely absent). I would like to advise you to read other COVID-19 papers published in PLOS ONE on the mental health of patients/general population, that can be used as examples.

We look forward to receiving your revised manuscript.

Kind regards,

Peter G. van der Velden, Ph.D.

Academic Editor

PLOS ONE

Reviewers' comments:

Reviewer's Responses to Questions

**Comments to the Author**

1. Is the manuscript technically sound, and do the data support the conclusions?

Reviewer #1: No

Reviewer #2: Partly

2. Has the statistical analysis been performed appropriately and rigorously? 

Reviewer #1: Yes

Reviewer #2: No

3. Have the authors made all data underlying the findings in their manuscript fully available?

Reviewer #1: Yes

Reviewer #2: No

4. Is the manuscript presented in an intelligible fashion and written in standard English?

Reviewer #1: Yes

Reviewer #2: No

5. Review Comments to the Author

Reviewer #1: This is an interesting study but it needs mayor revision before it can be considered again.

Authors should explain that they assessed non-hospitalized COVID.19 patients during the 14 days of quareenten. It is not clear. more data of COVID.19 fatures is needed. time from diagnosis is needeed. maybe anxiety and depression is not the same day one than 10 days after reclusing at home.

Differentiation between anciety and depresison in general population, hospitalized COVID-19 patients and long COVID patients is needed. Authors have assess moddo disorders in a manner that it is not commonly investigated in the literature, which increases the relevance of the paper, but more justification and discussion is needed.

What happened with these patients about 14 days? was any follow-up of them?

The role of gender needs extensive discussion since there is available data on the liteature for that

In conclusion, the paper needs xtensive inclusion of more data and extensive work before it can be reconsidered again.

Reviewer #2: Summary

The authors have attempted to assess the severity and prevalence of anxiety, depression, and stress and to identify risk factors in an adult cohort of participants with non-severe COVID-19 in the East Java province in Indonesia. They observed that depression was prevalent in 3.6% of the participants, anxiety in 14.3%, and stress in 7.9%. Furthermore, female gender was associated with a higher risk of anxiety and stress, and higher depression, anxiety, stress and total score. Based on these results, the authors conclude that depression, anxiety, and stress was prevalent in 3.6%, 14.3%, and 7.9%, that women are at higher risk and that women with confirmed COVID-19 should therefore be regularly evaluated.

Although this is an interesting study, a lot of work should be done prior to this work being ready to be published. My suggestions are described below:

General recommendations

- In my opinion, the level of English used could and should be better. There are quite many grammatical errors, suboptimal choices of verbs etc. Consider asking a native speaking colleague to copy-edit the manuscript.

- Including the outcomes of the univariate regression models does not fulfill the criteria of making all data underlying the findings described fully available; please add a dataset containing the raw data to fulfill this criterion.

Specific recommendations

Title

- I think “risk factor” should be “risk factors”, even though only one risk factor was identified.

Abstract

- I am missing the importance of your work/knowledge gap your research is filling in in your abstract

- “Study population was non-severe…” (line 31); I think “comprised” or “consisted of” would be a better verb

- “that was” (lines 31) should be “which were”

- “Collected data …. version 25.0” (lines 33-34) should be omitted from the abstract.

- “There were … in this study” (lines 34-35); Out of how many patients were those 608 included, in other words what was the inclusion/response rate.

- “From 608 respondents …” (line 35); in the phrase before it was already mentioned that there were 608 participants. I would suggest changing this to “Of these… “

- As you only used a screening instrument to detect psychological symptoms, I would suggest weaken the statements regarding having depression, anxiety, or stress. I think “reported” would be a better choice of verb.

- I would suggest adding the association between the severity of psychological symptoms (i.e., the scores) and female gender to your abstract.

- I don’t think the conclusion drawn can be extrapolated from the data presented. Although you demonstrate that female sex is a risk factor for psychological symptomatology, this does not mean that males doesn’t report any psychological symptoms (I imagine, as this comparison is not ready made in the manuscript).

Introduction

- I think in general the introduction is lacking important references.

- Lines 44 to 50 are not relevant, as everybody already knows about the COVID-19 pandemic. Try to be more creative when writing your introduction.

- I think much more literature regarding recent findings about psychological distress during COVID-19 should be added and described. The introduction in its present form lacks a description of why this research is important and what are the knowledge gaps. Why would you think that patients have psychological distress during COVID-19/a SARS-CoV-2 infection? What is already known? What isn’t known? What is your research going to add to the literature. I think “However, there is no study conducted that evaluates the mental health conditions among confirmed COVID-19 patients in Indonesia to this date” is not enough justification for your research.

- Overall, I think the introduction is to general, especially the first paragraph, and lacks a clear description of already published literature concerning this subject.

Methods

- “Study population was” (line 65): This is not correct English. Also, this sentence is too long, I would split it up in more sentences for clarity.

- “Inclusion criteria was … in this study” (lines 69-70): There are various grammatical errors in this sentence, please rephrase.

- “relevant institutional reviewer board” (lines 71-72): Please include the name of the institutional review board.

- I was expecting a paper describing the validity of the used screening questionnaire for reference 8. This reference however does not study this matter, but only uses the same instrument. Please provide a reference that is indeed studying the validity of the instrument. Are there any other references about the development of the DASS-21?

- Which sociodemographic characteristics were asked for? In the results, only age, sex, marital status, having a child and incomes are listed, are those also the only demographic variables that were gathered?

- I think the listing of the cut-off values is quite chaotic. Maybe it is an idea to add a table (in the main manuscript or additional files) which gives a clear oversight of the cut-of values.

- It is totally unclear to me why only the abovementioned variables were used to identify risk factors for the development of anxiety, depression, or stress. Was this based on existing literature? Please describe this in much more detail in the manuscript.

- Were all patients included in just one month (September 2020)? Otherwise, specify the start and ending date of the study.

- When and how were patients recruited and included?

- Why were patients with a history of mental illness excluded? How was this determined?

- You only describe that you use logistic and linear regression analysis, but it is not clear for which they are used. This should be made more clear.

Results

- I am missing the total number of patients who were eligible, from which the 608 participants were included. It might be an idea to include a flow diagram to show this.

- I am missing a table depicting the descriptives of the main outcomes, which are the prevalence and severity (i.e., sum scores) of the domains of the DISS-21. It is now only described in the text. In this way, you can omit the text regarding the distribution among mild, moderate or severe psychological distress, and just refer to the table.

- It might be an interesting idea to group the cohort based on their outcomes (i.e., patients who reported psychological stress vs. patients who did not), and compare these groups.

- I think the first paragraph could be written in a less point-to-point manner.

- Table S1: Where does ‘COR’ stand for? Abbreviations should be explaned for each table (not this is only done for table S2 and Table 3.

- Table 2: I do not understand why ‘having at least one child’ is not added to the multiple logistic regression model, as these variables had a p-value <0.25 in the univariate logistic regression model.

- Tabel 2: Why does this table include ‘depression’ as no multiple regression analysis was conducted for this outcome?

- Table 3: I would suggest adding all variables, including their Beta, SE, etc. for all components which were added to the model. In that way, you don’t have to describe everywhere for which other variables were adjusted.

- Table 3: I do not understand why the multiple linear model for anxiety was adjusted for having a child and marital status, when marital status did not have a p-value below 0.25 in the univariate analysis.

- Overall, I think the results section can be written in a more clear and organized manner.

Discussion

I think the discussion needs a lot of work; in its present form it is 1) way too shallow and 2) the authors have failed to discuss any truly important topics.

Try to rewrite the discussion using the following paragraphs:

First paragraph: Short repetition of the main results.

Second and further paragraphs: Discussing observed main results with recent literature.

Last paragraph: Limitations

In its present form, the authors already start discussing their results in the first paragraph by mentioning another study in COVID-19 confirmed cases. I think, if that is a point of discussion, the authors should elaborate more on these results, and also compare with, for instance, papers which report the prevalence of psychological distress after COVID-19. Other literature concerning this matter is hardly referred to.

Also, the only true discussion the authors describe is the fact the women are more prone to psychiatric sequelae after COVID-19. That is, in my opinion, not new or surprising at all and definitely not enough on its own for a discussion. The fact that the whole manuscript only counts 17 references, of which 5 are not even scientific papers, underscores my point.

Conclusion

The conclusion in its present form is just a repetition of the main results of this study and not so much of an conclusion, except that the psychological well-being of COVID-19 positive women should be evaluated regularly. In my opinion, the conclusion section should never contain number, as the results section is the only place to report these. As such, I think the conclusion should be rewritten.

6. PLOS authors have the option to publish the peer review history of their article (what does this mean?). If published, this will include your full peer review and any attached files.

Reviewer #1: No

Reviewer #2: No

---

## [Author Response · Author response to Decision Letter 0]

23 Feb 2022

Dear editor and reviewers,

Thank you for the comments and suggestions for this manuscript. We have uploaded the point-to-point response in the submission files.

---

## [Decision Letter · Decision Letter 1]

4 Apr 2022

PONE-D-21-21429R1Prevalence of and risk factors for depression, anxiety, and stress in confirmed non-severe COVID-19 patients in East Java province, IndonesiaPLOS ONE

Dear Dr. Salamah,

Thank you for submitting your manuscript to PLOS ONE. After careful consideration, we feel that it has merit but does not fully meet PLOS ONE’s publication criteria as it currently stands. Therefore, we invite you to submit a revised version of the manuscript that addresses the points raised during the review process. I agree with the comments of the reviewers (see below) and therefore will not repeat them here.

We look forward to receiving your revised manuscript.

Kind regards,

Peter G. van der Velden, Ph.D.

Academic Editor

PLOS ONE

Journal Requirements:

Reviewers' comments:

Reviewer's Responses to Questions

**Comments to the Author**

1. If the authors have adequately addressed your comments raised in a previous round of review and you feel that this manuscript is now acceptable for publication, you may indicate that here to bypass the “Comments to the Author” section, enter your conflict of interest statement in the “Confidential to Editor” section, and submit your "Accept" recommendation.

Reviewer #1: All comments have been addressed

Reviewer #2: All comments have been addressed

2. Is the manuscript technically sound, and do the data support the conclusions?

Reviewer #1: Yes

Reviewer #2: Yes

3. Has the statistical analysis been performed appropriately and rigorously? 

Reviewer #1: Yes

Reviewer #2: Yes

4. Have the authors made all data underlying the findings in their manuscript fully available?

Reviewer #1: Yes

Reviewer #2: Yes

5. Is the manuscript presented in an intelligible fashion and written in standard English?

Reviewer #1: Yes

Reviewer #2: Yes

6. Review Comments to the Author

Reviewer #1: Authors have answered most comment. The pper is much improved. Nevertheless, the introduction is still slightly confusing. Authorsd talk about mental health, but they do not clarify until the end of the introduction that they are talking of non infected patients. Authors should update current data on mental health in infected COVID-19 patients, both hospitalized and non-hospitalized. This should be clear in the introduction and use this information in the discussion

In addition, the title should use non-hospitalized patients instead of non-severe.

Reviewer #2: Dear authors,

Thank you for the opportunity to review your revised manuscript. I would like to congratulate you on your work as the revised manuscript has improved tremendously.

I however still think there should be made some minor revisions prior to the manuscript being ready for publication.

Abstract:

The authors did a good job improving their abstract and the use of the English language within the abstract. There are some small improvements to make:

1) Could you add the coefficients of your regression models when mentioning the risk factors, as only mentioning those won’t help the reader understand the correlation/association.

2) I would suggest adding some baseline demographics of the cohort, such as age, gender etc. to the abstract.

2) I would rephrase or even omit your recommendation. I think it is a nice conclusion that psychological distress was not that apparent in non-severe COVID-19 patients, but I do not believe that there should indeed be paid more attention to the psychological distress of these patients as psychological distress is that scarce. It seems even comparable or just a little bit higher (anxiety) than the expected prevalence of those disorders in non-COVID-19 times. Although I understand the authors tendency to give a recommendation, this recommendation is too general in my opinion, and I’m doubting whether the government is the one to pay attention, rather than employers (as HCW were more likely to report depression for instance).

Although it is the authors choice, I would prefer a structured abstract over a narrative abstract, especially considering the design of your study.

Introduction:

- Overall, I think the introduction improved a lot compared to the previous version. There are however some minor improvements to be made:

- I would pay some more attention to the first paragraph. The coronavirus disease 2019 was not identified as a global pandemic, but the spread of the virus causing this disease was classified as a pandemic. The disease does not infect people, the virus does. “Various preventive public health measures have been implemented”; as most of these measures have already been omitted, I would say ‘were implemented’ (have been implemented implies that it is currently still the case).

- There are still some grammatical errors in the introduction, especially in the use of verbs.

- I would suggest bundling the reasons for conducting your research (the lack of data from LMIC, differs across time periods etc.) to one paragraph. It helps you being more concise, which would improve the introduction.

Methods:

- I would like to congratulate the authors on the revisions made in the Methods, it has improved considerably. There are some minor changes to made:

- “… and information about informed consent was given before the consent form was signed.” I think this should be information about the study (I hope so at least)

- Thank you for summarizing the cut-offs of the questionnaire in a table, this is much more convenient for the reader and much less chaotic than the previous version. Good job.

- Although all possible risk factors are mentioned in the methods now, it is not really clear to me why you choose those variables. Was is based on literature concerning risk factors for psychological distress or were they chosen as they were easily gathered? Please explain in the methods. (in other words, why would you think that risk factors would be among those gathered variables?)

Results:

- Thank you for adding the flow diagram and the total number of patients assessed.

- “For regression analysis, all sociodemographic variables from Table 1 were included, except educational background and family income, as these variables had a higher percentage of missing data”; Could you also describe in the Methods section that variables that had a certain percentage of missing data were excluded from the data analysis.

Discussion:

Of all sections, the discussion section has improved the most. I think the discussion in its present form is very complete and most important topics are covered. I have no comments to improve the discussion.

Conclusion

The present conclusion is much better than the previous one. Only comment is that I would rephrase your recommendation, as also stated in the abstract.

7. PLOS authors have the option to publish the peer review history of their article (what does this mean?). If published, this will include your full peer review and any attached files.

Reviewer #1: No

Reviewer #2: **Yes: **Johan H. Vlake

---

## [Author Response · Author response to Decision Letter 1]

21 Apr 2022

Dear reviewers,

Thank you for the comments and suggestions for this manuscript. We have uploaded the point-to-point response in the online submission system.

---

## [Decision Letter · Decision Letter 2]

22 Jun 2022

Prevalence of and risk factors for depression, anxiety, and stress in non-hospitalized asymptomatic and mild COVID-19 patients in East Java province, Indonesia

PONE-D-21-21429R2

Dear Dr. Salamah,

We’re pleased to inform you that your manuscript has been judged scientifically suitable for publication and will be formally accepted for publication once it meets all outstanding technical requirements.

Kind regards and congratulations with your paper,

Peter G. van der Velden, Ph.D.

Academic Editor

PLOS ONE

Additional Editor Comments (optional):

Reviewers' comments:

Reviewer's Responses to Questions

**Comments to the Author**

1. If the authors have adequately addressed your comments raised in a previous round of review and you feel that this manuscript is now acceptable for publication, you may indicate that here to bypass the “Comments to the Author” section, enter your conflict of interest statement in the “Confidential to Editor” section, and submit your "Accept" recommendation.

Reviewer #1: All comments have been addressed

Reviewer #2: All comments have been addressed

2. Is the manuscript technically sound, and do the data support the conclusions?

Reviewer #1: Partly

Reviewer #2: Yes

3. Has the statistical analysis been performed appropriately and rigorously? 

Reviewer #1: Yes

Reviewer #2: Yes

4. Have the authors made all data underlying the findings in their manuscript fully available?

Reviewer #1: Yes

Reviewer #2: Yes

5. Is the manuscript presented in an intelligible fashion and written in standard English?

Reviewer #1: Yes

Reviewer #2: Yes

6. Review Comments to the Author

Reviewer #1: Authors have addressed all comments properly from this reviewer and the other reviewer. The paper can be now accepted

Reviewer #2: I would like to congratulate the authors on their revision, which considerably improved the manuscript.

I believe that it is now ready for publication.

7. PLOS authors have the option to publish the peer review history of their article (what does this mean?). If published, this will include your full peer review and any attached files.

Reviewer #1: No

Reviewer #2: **Yes: **Johan Hendrik Vlake

---

## [Editor Report · Acceptance letter]

27 Jun 2022

PONE-D-21-21429R2 

Prevalence of and risk factors for depression, anxiety, and stress in non-hospitalized asymptomatic and mild COVID-19 patients in East Java province, Indonesia 

Dear Dr. Salamah:

I'm pleased to inform you that your manuscript has been deemed suitable for publication in PLOS ONE. Congratulations! Your manuscript is now with our production department. 

Kind regards, 

on behalf of

Prof. dr. Peter G. van der Velden 

Academic Editor

PLOS ONE